# The First Comprehensive Examination of Male Morphometrics and the Operational Sex Ratio of the Black Sea Turtle (*Chelonia mydas agassizii*) Population in Colola, Michoacan, Mexico

**DOI:** 10.3390/ani15010002

**Published:** 2024-12-24

**Authors:** Carlos Delgado-Trejo, Miguel Ángel Reyes-López, David Guillermo Pérez-Ishiwara, Ricardo Lemus-Fernández, Fátima Yedith Camacho-Sánchez, Cutzi Bedolla-Ochoa

**Affiliations:** 1Centro de Biotecnología Genómica, Instituto Politécnico Nacional, Reynosa 88710, Tamaulipas, Mexico; mreyesl@ipn.mx; 2Instituto de Investigaciones Sobre los Recursos Naturales, Universidad Michoacana de San Nicolás de Hidalgo, Morelia 58330, Michoacán, Mexico; 3Escuela Nacional de Medicina y Homeopatía, Instituto Politécnico Nacional, Ciudad de México 07320, Mexico; dperez@ipn.mx; 4Facultad de Biología, Universidad Michoacana de San Nicolás de Hidalgo, Morelia 58000, Michoacán, Mexico; ricardolemus020272@gmail.com; 5Unidad Académica Multidiciplinaria Reynosa-Aztlán, Universidad Autónoma de Tamaulipas, Reynosa 88740, Tamaulipas, Mexico; fatimaycs@gmail.com

**Keywords:** black sea turtle males, operational sex ratio, morphometrics

## Abstract

A comprehensive analysis was conducted on 10 morphometric traits of breeding male black sea turtles from the population in Colola, Michoacán, Mexico. The results indicate that these males have a smaller body size than other populations of green sea turtles in their pantropical distribution range. On average, male black sea turtles are 15.2 cm smaller than nesting females, which may have implications for their sexual maturity and mating success. The operational sex ratio (OSR) reflects the number of males available for mating with receptive females. In this study, an OSR of 2.9 males for every 1 female in a multi-year analysis of the Michoacan population was found. We also identified monthly variations in OSR values at the beginning and end of the nesting season, likely due to the differential arrival of males to the breeding area and the varying number of sexually receptive females. Research on this subject is scarce, and this is because reproductive interactions (mate and courtship) take place far from the nesting area. This distance complicates their study due to accessibility and cost issues.

## 1. Introduction

Over millions of years of evolution, sea turtle species have developed morphological adaptations that are essential for survival in the marine environment [1,2]. Some of the main adaptations are the loss of the articulations of the extremities, which have been modified in the form of fins to facilitate movement [3], modification of the lacrimal glands to eliminate excess salts from the body fluids absorbed from seawater, modification of the structure and shape of the carapace in a hydrodynamic way, and the significant reduction in bone material [4].

The size and shape of the body are the most conspicuous and fundamental characteristics in the life history of an organism; these features reflect evolutionary characteristics that intervene in the success of the species, populations, and individuals, as well as being important indicators of the morphological adaptations of a species to its environment [5,6,7,8,9,10,11,12]. In a concrete sense, the physical characteristics of the individuals influence the structure of the population, and intervene and maintain its balance in aspects such as feeding, growth, association, reproduction, and intraspecific interactions, among others. This, in turn, influences distribution at local and regional scales, which in the long term is reflected in evolutionary processes and facilitates variation in diversity and speciation [13,14,15,16]. The observed morphological variation can be separated into its environmental and genetic components [17,18,19,20].

All sea turtle species have evolved from a common ancestor [1]. As a result, their physical adaptations to the environment are similar across all species. The primary differences between them are external characteristics, such as size, coloration, conformation, and number of plates and scales, among others [21,22,23,24].

The size of sea turtles can be the result of both biotic and abiotic factors, such as the condition of the parents [25,26,27,28], incubation conditions, and food resources during growth [29,30,31,32,33].

The main morphometric feature taken as a reference for descriptions is the size of the carapace, and this has been used to establish different parameters including growth rate, the link between size and fecundity, and lineage [12,34,35,36,37].

The measurement methods are mainly based on those proposed by Frazer [38], Pritchard et al. [39,40,41], and Bolten [42], and eight main characters are taken into account: straight carapace length (SCL), curved carapace length (CCL), straight carapace width (SCW), curved carapace width (CCW), plastron length (PL), infra-anal scale length (ISL), head width (HW), and body height (BH). Several studies have been conducted to morphometrically describe the east Pacific green sea turtle (also known as black sea turtle (*Chelonia mydas agassizii*). However, the descriptions have been made almost entirely of adult females and recently hatched hatchlings, leaving somewhat behind the knowledge of the characteristics of adult males. In part, this absence of information is due to the difficulties involved in obtaining this information because the males remain in the sea all of the time after they emerge from the nest [43].

Another important aspect that also determines population structure is the operational sex ratio (OSR), which indicates the ratio of sexually available males to sexually available females in a population [44,45,46]. The relative proportions of breeding males and breeding females in a population influence reproductive dynamics, including intersexual selection and intrasexual competition [47,48]. The theory of sexual selection indicates that the operational sexual ratio (OSR) is a main indicator of the opportunity for sexual selection. A bias in the OSR predicts that the sex with the highest percentage will be the dominant competitor for access to mating partners. Thus, the unbiased OSR of 50% is of special interest because it is around this point that sex roles are predicted to be ideal (the ratio should be a 1:1 ratio of males to females). The more biased the OSR, the more intense the competition, and the stronger the sexual selection [49]; if certain individuals of this sex have a heritable trait that consistently favors them for mating competition, this trait will be sexually selected.

If there are more females than males available for mating in the breeding population (i.e., the operative sex ratio, OSR, is female-biased), males can afford to be choosy. In the pipefish (*Syngnathus typhle*), females compete for males, which are very selective [50]. In nature, OSRs are typically female-biased but may occasionally be male-biased.

It is well known that worldwide and for different species, the sex ratio of sea turtle hatchlings is biased towards females [51]. Sexual selection results from differences in mating success [52] and its success covaries with the phenotypic expression of each character [53].

In an OSR investigation in *Drosophila melanogaster*, it was observed that the greatest male mating advantage is seen with sex-related change. Larger males have been more successful in mating when the sex ratio was equal or when males were more abundant and therefore in competitive conditions. Male size also influenced the mating order, with larger males mating before smaller males [54].

Berry and Shine [55] found that the mating strategies of aquatic turtles appeared to involve an elaborate courtship, suggesting female choice, and almost universally showed sexual dimorphism in which males were smaller than females. In the case of terrestrial and semi-aquatic turtles, they showed larger males than females of the same species, and suggested the existence of combat between breeding males and even forced insemination behavior in the case of semi-aquatic species. In sea turtles, secondary sexual characteristics are marked: reproductive males have a longer prehensile tail with terminal claw, larger and recurved claws on the front flippers [56], and a soft and more concave plastron [57]. For green turtles, Godley et al. [58] found a clear and consistent pattern of sexual dimorphism, with males being the smaller sex.

Therefore, data regarding the body size and sex of the animals are indispensable for understanding the structure and demography of the populations. For male black sea turtles, there is little information on morphometric aspects and operational sex ratio, which are referred to in brief descriptions, but no specific and detailed studies have been carried out.

This study aimed to analyze morphometric characteristics of reproductive males and the operational sex ratio of the eastern Pacific black sea turtle population, referred to as the black sea turtle, along the coast of Colola, Michoacan, Mexico.

## 2. Materials and Methods

To determine the morphometric characteristics and the operational sex ratio (OSR) of the male black sea turtle population in Colola, Mexico (18°30′0″–18°0′0″ N, 103°40′0″–102°50′0″ W) (Figure 1), morphometric data were recorded during the 2004 and 2018 seasons and observations of the reproductive interactions of individuals were made during four breeding seasons (2004, 2009, 2017, and 2023).

The morphometric characterization of the black sea turtle reproductive male population was obtained considering the criteria established by Pritchard et al. [39] by measuring the following 10 morphometric variables: straight carapace length (SCL), curved carapace length (CCL), straight carapace width (SCW), curved carapace width (CCW), plastron length (PL), head width (HW), total tail length (TTL), and body height (BH), Length of anterior nails (LAN), length of posterior nails (LPN), and weight of 132 males (n = 100 males in 2004 and n = 32 males in 2018) at Colola Beach, Michoacan, Mexico. The capture of the individuals was carried out using the “swimming” capture technique, which consists of approaching mating males from a boat with an outboard motor from approximately 30 m away, then swimming to approach the males to capture and take them on board for measurements. Body lengths were registered using a flexible tape measure for CCL and a caliper for SCL (0–150 cm). To ensure that sampling was not duplicated in the same individuals, males were marked with Inconel steel tags on the left posterior flipper.

Statistical analyses of morphometric variables were conducted using descriptive statistics using the statistical package SYSTAT 12.0.

The OSR was assessed through the methods of focal behavioral sampling of individuals, ad libitum, and the continuous behavioral recording method [59]. Records were made of the courtship and mating interactions of breeding adult black sea turtles from a 40 m high natural rock platform located at the west end of the beach in front of the sea (Figure 2). From this natural platform, it was possible to make observations of reproductive interactions during the four breeding seasons. Data were meticulously collected through direct observations and extensive photo and video recordings. Equipment used included a Nikon D90 DSLR with a 150–600 mm zoom lens (Sigma Corporation, Kawasaki, Kanagawa, Japan), binoculars, a Bushnell 36x monocular (Overlandpark, KS, USA), a Canon HD CMOS Pro XA11 video camera (Tokio, Japan), and a DJI Mini 3 drone (Shenzhen, Guangdong, China). This robust set of devices allowed for the accurate capture of images and videos documenting breeding interactions.

Sampling of adults in these interactions was conducted from 08:00 to 18:00 h if weather conditions permitted. OSR assessments were conducted from September through December when courtship and copulation interactions are most frequent and abundant in the area. The nesting season at Colola beach begins in August and ends in April; however, nesting is common during the warmest season of the year (May–July).

## 3. Results

### 3.1. Morphometric Characterization

The descriptive statistics of the metric variables of male black sea turtles (n = 132) are shown in Table 1.

Regarding the results for the weight of the sampled individuals of black sea turtles, a mean of 45.7 ± 6.5 kg (range 30–61 kg) was obtained.

### 3.2. Operational Sex Ratio

A total of 336 h of observations of black sea turtle courtship and copulation activities were conducted at Colola beach during the four study seasons. Six hundred and fifty-three mating groups were observed, involving 1986 males and 669 females. An estimated overall operational sex ratio (OSR) of 2.96:1 males per female (≅3.0:1 male/female), with a range of 1–17 males per female in mating groups, was estimated (Figure 3).

By sampling period, we obtained different OSR estimates. In 2004, 150 h of observations were carried out in which a total of 352 copulation groups were identified, involving 1079 males and 368 females, with an estimated OSR of 3.0:1 male/female (range 1–13 male/female). In this period, the monthly OSR estimates showed slight variations: in September, the estimated OSR was 2.8:1; in October–November, it was 3.1:1 and in December, the estimated PSO was 3.3:1 male/female.

In the 2009 breeding season, 86 h of observations were made, and 83 copulation groups were recorded, involving 442 individuals: 359 males and 86 females. The highest estimate of OSR was obtained in this period, with 4.3:1 male/female (range 1–17 males per female). In the copulation groups recorded, it was observed that in 26.5%, only one male participated, while in 73% of the copulation groups, two or more males were observed.

In the 2017 season, 40 h of mating and courtship observations were conducted. Forty mating groups were detected in which 163 individuals (123 males and 40 females) were identified (range 1–9 males per female), and an OSR of 3:1 male/female (range 1–9 males per female) was estimated. During this period, only one male was observed in 17.5% of the mating groups, while two or more participating males were observed in 82.5% of the groups.

In the breeding season of 2023, 60 h of observations of reproductive interactions were carried out in which 178 mating groups were identified. A total of 603 breeding adults (425 males and 178 females) were identified, with an estimated OSR of 2.3:1 male/female (range 1–12 males per female). This was the lowest OSR estimate of the entire sampling period considered in this work. On the other hand, in 41.5% of the mating groups, only one male was observed, while in 58% of the groups, two or more males were observed.

Of the 653 copulation groups observed in this work, in 34.3% of these groups, only one male was observed in reproductive interactions with a female, while in 65.7% of the groups, 2–17 males were observed in reproductive interactions with a female (Figure 4).

## 4. Discussion

In this work, we present 10 morphometric measurements of male black sea turtle breeders. The average length of males was 67.5 cm ± 3.5 (SCL), while the average curved carapace length (CCL) was 70.5 cm ± 5.3; straight carapace width was 54.9 cm ± 4.3 (SCW) and total tail length (TTL) was 42.0 cm ± 2.9. Morphometric analysis of the male black sea turtles in this work showed that they are the smallest compared to the size of males reported in other populations of Chelonia by Hirt [60]. On the other hand, black turtle males are, on average, 15.2 cm smaller than females (85.7 cm CCL, ref. [12]) of the same population, which has important implications for the age of the sexual maturity of males and the mating process. Differences in the size of females and males have been reported in the breeding population of green turtles at Aldabra Atoll, Seychelles; females were larger than males (CCL, 112 cm ± 4.9 and 102.6 cm ± 4.6, respectively) [61]. This analysis of 10 morphometric measurements of male black sea turtles allows us to affirm that males are smaller than those of other Chelonian populations and are only similar in size to male black sea turtles reported in the Galapagos Islands [62]. Differences in the sizes of males from Michoacán and Galapagos with other populations may be due to the fact that males residing in other feeding areas have different growth rates, because these are highly dependent on the abundance, acceptance, and nutritional quality of the food [63,64,65].

The most notable characteristics of males, besides their small size (CCL), are the size of the tail and the size of the front and rear flipper nails. It appears that body size is adapted to the conditions of the eastern Pacific. Environmental variations, potentially influenced by the El Niño (ENSO) phenomenon, have played a significant role in shaping the life history traits of black sea turtle populations in this region, including the population found in Michoacan [12]. Male tail size may be a morphometric characteristic that can indicate sexual maturity, since all captured males were involved in copulation groups either as satellite/escort males or mates, so males with a tail size below the minimum size reported in this work (31 cm TTL) would correspond to sexually immature males. Some studies have reported the size of male and female breeding sea turtles, and in some species, there is a consistent finding that adult males are smaller than females [1,58,66].

For sea turtles, not much is known about the relative age at which males and females reach sexual maturity [12,67]. In theory, when sexual dimorphism exists and adult females are larger than adult males, if growth rates are similar between the sexes, males would be expected to reach sexual maturity earlier. Therefore, it could be suggested that, in males, reproductive success is influenced by aspects related to behavior, disposition, energy, and movement capacity, rather than by size.

### Operational Sex Ratio

There is little information on estimates of operational sex ratio in sea turtles obtained through direct observations of reproductive interactions. Most estimates of OSR are based on multiple paternity analyses obtained from molecular biology tools.

Some estimates of OSR in the black sea turtle population have been reported since conservation activities began in 1978 in the Michoacan population. Cliffton and Cornejo [68] reported an OSR of 1.04:1 male/female in 362 mating groups observed in aerial surveys, while Alvarado and Figueroa [66] reported an OSR value of 2.5:1 male/female, with a range of 1–6 males per female.

Both estimates were made during the decline in the population. These low OSR estimates may be the result of the low population of breeding males and females during the legal harvest of breeding adults in the 1970s and 1980s in the Mexican Pacific. The OSR estimate obtained in this work (2.96:1 male/female) coincides with the report of multiple paternity found in black turtle females at Colola Beach by Lara-De la Cruz et al. [69] who report the participation of three paternal genotypes in the fertilization of black turtle nests.

In the 2004 breeding season, OSR estimates vary monthly, increasing from September (2.8:1) to December with 3.3:1 male/female. This temporal variation in OSR is likely the result of the gradual arrival of males to the breeding area of Colola Beach, with the number of males increasing in December when the peak of female black sea turtle nesting occurs. At the beginning of the nesting season in September, female receptivity is likely to be lower. This reduction may discourage males from attempting to copulate with females. Additionally, the low density of breeding adults could result in fewer encounters between males and females.

It is important to mention that due to the high concentration of males in front of Colola and the reproductive behavior of the males, which try to maximize their reproductive success by increasing the number of copulations they manage to have, it is likely that some males participate in different copulation groups and are counted repeatedly, particularly those that do not copulate frequently but participate as escort or satellite males, which, although unable to access copulation, attempt to maximize their reproductive success by preventing other males from copulating more frequently through copulation interference [70]. However, because copulation groups are distributed along 5 km off Colola and up to 500 m away from the beach, the probability of counting the same males is reduced and does not affect the OSR estimation. Other authors mention that males may adopt courtship strategies with less energetic expenditure, and although these males may not search as large an area, they will be able to actively participate in mate searching and mate acquisition for longer. Courtship aggregations may show a significant intra- and interannual variation density and ratio of breeding males and receptive females [71].

In the 2023 season, the lowest estimate of OSR was obtained, of only 2.3:1 male/female. This evaluation was obtained under very particular conditions that were not observed in the other sampling periods. During the entire season (September–December) there were very few copulation groups, particularly in December when only eight groups were observed. Low male reproductive activity may have been influenced by conditions created by the El Niño phenomenon (ENSO) in the Mexican Pacific.

Limpus and Nichols [72,73] mention that, at least in green turtles (*C. mydas*), reproductive rates in females and males are regulated by regional climatic events generated by the El Niño phenomenon, and it appears that levels of endogenous energy reserves may play a vital role in intra- and interannual reproductive effort in both sexes.

Therefore, it is suggested that the high availability of males in copulation groups may translate into a high frequency of multiple paternity in black sea turtle nests. Chassin et al. [74] report a frequency of multiple paternity of 75% in black sea turtle hatchlings at Colola Beach, Michoacan.

## 5. Conclusions

This paper reports, for the first time, a complete analysis of the morphometrics of breeding males of the eastern Pacific green turtle, known as the black sea turtle (*Chelonia mydas agassizii*).

These males are particularly small compared to other populations of male green sea turtles reported in the literature.

Breeding black turtle males are, on average, 15.2 cm CCL smaller than females. This small size of males could explain the characteristic shape of the female carapace, which has a particular narrowing at the back of the carapace, and which would allow the smaller males to hook onto larger females with their anterior and posterior fins and their long nails.

The difference in size between males and females would have consequences for the age of sexual maturity of males, who would mature at an earlier age than females.

Regarding the operational sex ratio, we present a multiannual analysis of the OSR values of the black sea turtle population in Michoacan, which allows us to understand an aspect of the reproductive ecology of sea turtles about which there is scarce information. This is because copulation and courtship interactions take place at sea, making it difficult to study for most sea turtle populations, especially those with pelagic habits in the adult stage.

Adult black sea turtle breeders exhibit polyandrous behavior in which females copulate with different males throughout the breeding season, mainly from September to December. This behavior provides opportunities for females to select different males to increase the fitness of the hatchlings produced. Since there is no intrasexual selection for the establishment of dominance among breeding males, females discriminate among a good number of available males, i.e., the males they consider the most suitable through the identification of morphological and behavioral traits (sexual courtship), which we still do not know. However, the morphometric traits reported in this work seem to meet the morphometric conditions selected by nesting females. Nonetheless, aspects related to the behavior of males during courtship remain to be elucidated.

## Figures and Tables

**Figure 1 animals-15-00002-f001:**
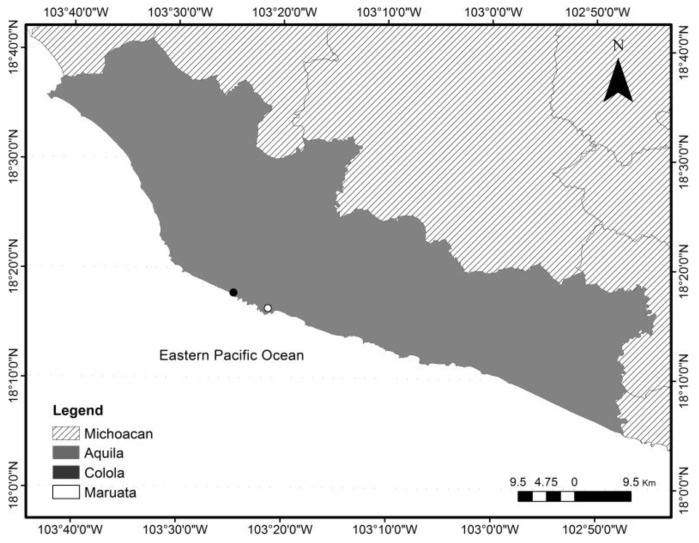
Black sea turtle (*Chelonia mydas agassizii*) breeding area in Colola, Mexico.

**Figure 2 animals-15-00002-f002:**
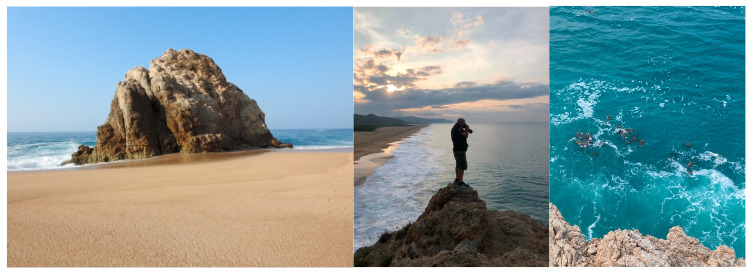
A natural platform for observation of black sea turtles of mating and courtship interactions.

**Figure 3 animals-15-00002-f003:**
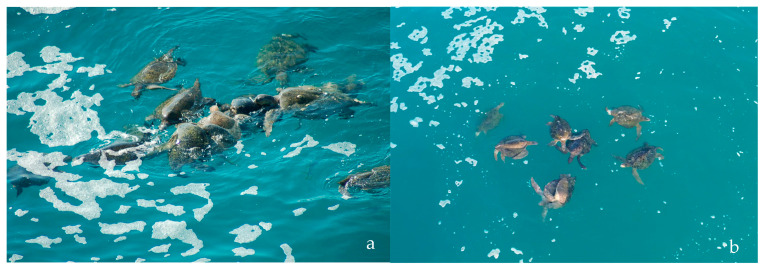
Mating groups of black sea turtle breeding adults. (**a**) Image obtained with a reflex camera from the observation platform. (**b**) Image obtained through drone survey.

**Figure 4 animals-15-00002-f004:**
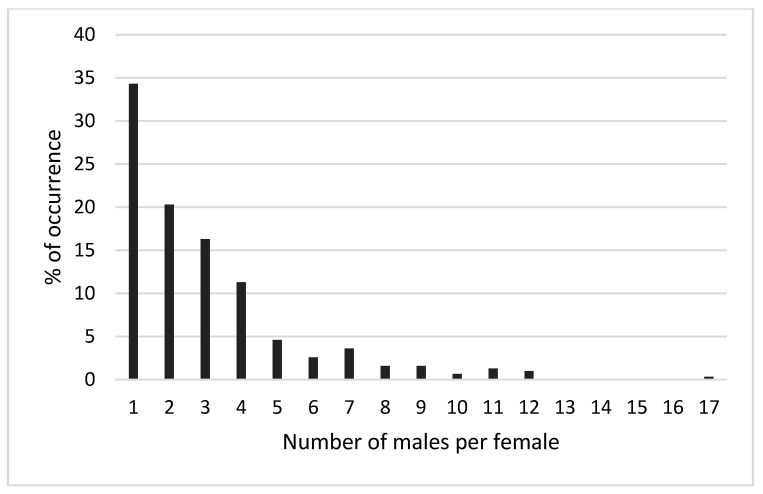
Percentage of number of males per mating group.

**Table 1 animals-15-00002-t001:** Measures of central tendency of 10 morphometric variables of adult male black turtles (measurements in cm).

Variables	Max.	Min.	Mean	S. Desv.	CV
SCL	76.6	61.1	67.5	3.5	4.7
CCL	80.1	63.4	70.5	5.3	4.9
SCW	75.0	50.0	54.9	4.3	7.8
CCW	80.0	62.0	70.6	3.8	5.4
PL	66.0	43.0	45.2	6.8	15.0
HW	12.0	8.9	9.5	1.2	12.6
TTL	56.0	31.0	42.0	2.9	6.0
LAN	3.0	1.7	1.8	0.21	11.6
LPN	1.8	0.9	1.0	0.15	8.3
BH	34.0	21.0	25.3	2.9	11.4

Straight carapace length (SCL), curved carapace length (CCL), straight carapace width (SCW), curved carapace width (CCW), plastron length (PL), head width (HW), total tail length (TTL), length of anterior nails (LAN), length of posterior nails (LPN), and body height (BH).

## Data Availability

The data presented in this study are available on request from the corresponding author.

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
