# Peer review of "The First Comprehensive Examination of Male Morphometrics and the Operational Sex Ratio of the Black Sea Turtle (Chelonia mydas agassizii) Population in Colola, Michoacan, Mexico"

_animals, 2024, doi:10.3390/ani15010002_

Round 1

Reviewer 1 Report

Comments and Suggestions for Authors

Dear Author(s),

Morphological data and OSR provide important information about the male black sea turtle. However, I would like to draw attention to a point that you should clarify, especially in the MM section. What is the recurrence rate of males that you observe in copulation groups? Perhaps you observed the same males. Please provide more descriptive information and discuss these limitations of your study. There are also some spelling and presentation errors in the introduction and results section. I present these and all other suggestions in the PDf file.

Best

Author Response

Comments 1:  Morphological data and OSR provide important information about the male black sea turtle. However, I would like to draw attention to a point that you should clarify, especially in the MM section. What is the recurrence rate of males that you observe in copulation groups? Perhaps you observed the same males. Please provide more descriptive information and discuss these limitations of your study. There are also some spelling and presentation errors in the introduction and results section. I present these and all other suggestions in the PDf file.  

Response 1: We agree with this comment. Therefore, during observations of copulation groups it is indeed possible to observe the same males in different copulation groups, however, we can only identify males that have some particular characteristic that distinguishes them, for example, some scars on the tail, carapace and fins; others have some epibionts such as balanos (Chelonibia testudinata) attached to the carapace that allow us to distinguish them in the observations.

However, as copulation groups occur relatively close to each other, males that are not copulating attempt to engage in copulation interference to get the copulating male and female to desist through nibbling on the tail, fins and neck.

Other males, when unsuccessful, simply seek out other females and form part of another copulation group, so it is possible to observe the same males in different copulation groups, although in some cases they may be copulating or escorting males. We believe that this does not affect OSR estimates.

Males try to maximise their reproductive success by copulating with as many females as possible, however, the polyandrous system exhibited by females does not allow for patterns of male dominance to be established, i.e. the recurrence of any particular male copulating, although not estimated in this work, is believed to be low.

Reviewer 2 Report

Comments and Suggestions for Authors

Dear authors,

I found your manuscript very interesting. However, you should improve some little aspects to offer the reader a more up-to-date perspective.

In the Introduction, no mention is reported about gonad development and post-mortem analysis. Please, integrate with some text and references.

You should also include the IUCN red list ranking of the species in the Introduction within some conservation perspectives that your study could contribute to. Moreover, please, describe better the species and its distribution which are interesting points for the reader who doesn't know this species.

In the Materials & Methods, please, provide an overview of the study area, why it was selected, and how this population is monitored and protected. Furthermore, you should include a mention of the characteristics of the mating behavior: how did you decide if that behavior was mating? If you can, please, include a reference to a sea turtle ethogram about this topic. 

The same conservation perspective should be discussed in the Discussion and Conclusion chapter. A work about sea turtles without a conservation perspective sounds quite anachronistic and just a simple data collection without any practical application or contribution to the protection of this species.

In the Dicsussion, you should also include the limit of your work as it is based only on captured animals and mating observation, and no post-mortem investigations on gonad development were conducted.

Many thanks

Author Response

Comments 1: I found your manuscript very interesting. However, you should improve some little aspects to offer the reader a more up-to-date perspective.

Response 1: We agree with this comment. We made changes to improve the manuscript

Comments 2: In the Introduction, no mention is reported about gonad development and post-mortem analysis. Please, integrate with some text and references.

Response 2: Due to the fact that the work was with adult breeding individuals, the analysis of post-mortem gonad development was not necessary, because females and males presented their distinctive secondary sexual traits that identified them as adult breeders (males with long tails) and females with very short tails in relation to males.

Comments 3: You should also include the IUCN red list ranking of the species in the Introduction within some conservation perspectives that your study could contribute to. Moreover, please, describe better the species and its distribution which are interesting points for the reader who doesn't know this species.

Response 3: Thank you for your suggestion, we have already included your suggestion in the manuscript.

Comments 4: In the Materials & Methods, please, provide an overview of the study area, why it was selected, and how this population is monitored and protected. Furthermore, you should include a mention of the characteristics of the mating behavior: how did you decide if that behavior was mating? If you can, please, include a reference to a sea turtle ethogram about this topic.

Response 4: Thanks for the suggestion, it has been included in the manuscript.

It is very easy to distinguish copulation behaviour, the male remains anchored on the female with his tail wrapped around the female's tail, this behaviour can last on average up to 3 hrs. Although sperm transfer may take place 5 to 10 min after mating, the male remains attached to the female to prevent other males from copulating with her and sperm competition from occurring.

Comments 5: The same conservation perspective should be discussed in the Discussion and Conclusion chapter. A work about sea turtles without a conservation perspective sounds quite anachronistic and just a simple data collection without any practical application or contribution to the protection of this species.

Response 5: We agree with the comment, from a conservation point of view, we believe that the OSR can be an indicator of the number of breeding males in the population, during the period of decline of this population the OSR was estimated at 2:1 male/female and now with the recovery of the population we have found OSR above 3 males per female. The increase in males copulating translates into an increase in the Darwinian fitness of the offspring through an increase in the frequency of multiple paternity of the descendants.

Comments 6: In the Discussion, you should also include the limit of your work as it is based only on captured animals and mating observation, and no post-mortem investigations on gonad development were conducted.

Response 6: As mentioned in answer 1, during this work we were able to confidently identify females and males in the copulation groups and because all individuals were in reproductive interactions, we are almost 100% certain that females and males were sexually mature. Analysis of post-mortem gonad development was not considered.

Round 2

Reviewer 1 Report

Comments and Suggestions for Authors

Dear Author(s),

Following the revision, MS is now a more suitable choice. I would like to make a minor suggestion: when you are defining the distinction between male and female in Drosophila melanogaster (which is an invertebrate), it would be beneficial to also include the differences from higher organisms or even sea turtles. In the previous version, I proposed that you write a few sentences about this topic. Please make a brief note about sea turtles as well as higher organisms.

Best

Author Response

 Comment 1 Following the revision, MS is now a more suitable choice. I would like to make a minor suggestion: when you are defining the distinction between male and female in Drosophila melanogaster (which is an invertebrate), it would be beneficial to also include the differences from higher organisms or even sea turtles. In the previous version, I proposed that you write a few sentences about this topic. Please make a brief note about sea turtles as well as higher organisms.

Response 1 We agree with the comment, in round 1 we did not have the bibliography suggested by the reviewer. once revised in this second round, we included in the manuscript information recommended by the reviewer on sexual dimorphism in sea turtles and the strategies used in groups of terrestrial, semi-aquatic and marine turtles as megafauna.
